# Mercury in Selected Abiotic and Biotic Elements in Two Lakes in Poland: Implications for Environmental Protection and Food Safety

**DOI:** 10.3390/ani13040697

**Published:** 2023-02-16

**Authors:** Monika Rajkowska-Myśliwiec, Mikołaj Protasowicki

**Affiliations:** Department of Toxicology, Dairy Technology and Food Storage, Faculty of Food Science and Fisheries, West Pomeranian University of Technology in Szczecin, 71-459 Szczecin, Poland

**Keywords:** mercury, methylmercury, bioaccumulation, biomagnification, fish organs, common reed, water, bottom sediments, dietary exposure, risk assessment

## Abstract

**Simple Summary:**

This study assesses the state of mercury pollution in the ecosystems of two studied lakes and indicates which species of fish are safe for consumption by determining mercury levels in selected elements of the lake environment. It also examines the impact of selected biotic (length and weight of fish) and abiotic (Hg content, pH, O_2_ in water and sediment, water transparency) factors on the ability of fauna and flora to bioaccumulate mercury. Based on the maximum residue levels (MRLs), the muscles of pike, bream and roach from both lakes were found to be safe for consumption. No significant health risk was identified based on estimated daily intake (EDI), target hazard quotient (THQ) and tolerable weekly intake (TWI) values.

**Abstract:**

Mercury, which tends to bioaccumulate and biomagnify in aquatic food webs, poses a potential health risk to wildlife and to consumers of predatory fish in particular. Its concentration in biota can be high even at low environmental concentrations. Therefore, the aim of this study was to determine mercury in both abiotic (water and sediment) and biotic elements (common reed (*Phragmites australis*) and fish: pike (*Esox lucius*), bream (*Abramis brama*) and roach (*Rutilus rutilus*)) in the context of assessing the pollution of two lakes in Poland and the safety of fish consumers. The possibility of Hg biomagnification in fish was also considered. Mercury was determined by means of cold vapor atomic absorption spectrometry (CVAAS). The concentrations of Hg in water and bottom sediments of Lake Ińsko were lower than in Lake Wisola. In the bottom sediments of both lakes, a positive correlation was found between the Hg content and organic matter. The concentration of mercury in the organs of common reed did not exceed 0.017 mg/kg dry weight (dw), and its distribution can be presented as follows: root > leaves > stems > rhizomes. In fish organs from both lakes, the average mercury content did not exceed 0.086 mg/kg of wet weight (ww) and in most cases it was the highest in pike. Higher values were only observed in the muscles and skin of roach. This indicates a lack of biomagnification in the relationships between planktivorous-predatory and benthivores-predatory fish. Based on the maximum levels of mercury in fish and the calculated parameters, i.e., estimated daily intake (EDI), target hazard quotient (THQ) and tolerable weekly intake (TWI), the muscles of the examined fish were found to be safe for consumption. The average dietary exposure to total mercury (THg) and methylmercury (MeHg) was below 0.3% of the TWI.

## 1. Introduction

Mercury can accumulate to high levels in water, sediments and aquatic organisms such as benthos and fish, and can be highly toxic to people and animals [1]. It is a persistent element that can be mobilized from natural deposits into the biosphere through anthropogenic activities or natural processes [2]. In some cases, heavy metal concentrations can be relatively low within the water column, but higher in the sediment [3]. It has been reported that far less than 1% of potentially toxic elements (PTEs) remain dissolved in water, and therefore water quality is only a temporary reflection of environmental contamination status [4]. Hence, as much as 99% of trace metals are stored in sediments, which makes them the major sinks and carriers of contaminants in aquatic environments [5]. However, metals can pass from bottom sediments into water when water pH is low [6] or when reducing conditions are created there.

Mercury can undergo methylation into methylmercury (MeHg; CH_3_Hg^+^). In this form, it can biomagnify and bioaccumulate in food chains to levels that can be dangerous to humans [2,7]. The toxicity of methylmercury is of great concern to wildlife and human populations due to its neurotoxic and endocrine disrupting capabilities [8]. It has many detrimental effects on human health, most of them neurological, such as distal sensory disturbances, constriction of visual fields, loss of muscle control, dysarthria (speech disorder), auditory disturbances and tremors [9]. There is increasing evidence that even low-level exposure to MeHg can have detrimental effects on cardiovascular health in adults and neurological development in fetuses and young children [10].

Studies of water and bottom sediments do not fully explain the fate of mercury in the environment. According to critical review, by Urlich et al. [11] mercury, particularly methylmercury, is effectively taken up by aquatic biota, and bioconcentration factors in the order of 10^4^ to 10^7^ have been reported. Mercury accumulation in the aquatic food chain therefore can be high even when environmental concentrations are generally very low. Thus, only studies based on living organisms can give a picture of the threat resulting from bioavailability of pollutants present in the aquatic environment. Both plant and animal organisms can serve as good indicators of the state of the environment. Among macrophytes, the common reed (*Phragmites australis*) is recommended for testing water pollution with heavy metals [12,13,14].

Common reed is a widely spread rush species that grows in lakes. Although naturally occurring plants play an important role in mercury biomagnification in food chains, little research has been performed on the effects of mercury on natural vegetation, particularly regarding THg and MMHg [15,16]. Due to its fibrous roots and their large contact areas, as well as its production of large amounts of aboveground biomass, the common reed has been found to be very efficient in the accumulation of heavy metals [17]. This plant may therefore be used as a biomonitor of water and sediment contamination with trace elements, including Hg [18].

In addition, due to their feeding practices and living in the aquatic environment, fish are both particularly sensitive and highly exposed to pollution. They have been also found to be good indicators of water contamination in aquatic systems because they occupy different trophic levels, are of different sizes and ages and, compared to invertebrates, are more sensitive to many toxicants. Moreover, fish are a convenient test subject for indication of ecosystem health [19,20]. Pikes are top predators that occupy higher trophic levels than bream, benthophages and roaches, planktonophages. Due to its predatory lifestyle, the pike contributes to shaping the fish stock of the reservoir. Including an appropriate pike stock for the capacity of the reservoir has a positive effect on the fish stock [21]. Many authors [22,23] have considered bream to have a negative impact on water quality in reservoirs, and have linked the species with eutrophication processes. The bream’s benthivorous and/or planktivorous feeding habits are frequently a cause of bioturbation, which can increase nutrient release from the reservoir bottom [24]. Roaches are classified as herbivorous fish, i.e., those that in natural conditions take food of plant origin, although it also eats large amounts of animal food. The roach is very important as a food for piscivorous fish and it is hence an important link in the transfer of Hg within the food web [25]. In many lakes, especially the bream type, the diet of older pikes is dominated by roach. However, this species competes with bream for food [20].

An analysis of fish from different trophic levels can provide information about the bioaccumulation and biomagnification that occurs even if the different species do not prey directly on each other [26]. The bioaccumulation of metal by fish and its subsequent distribution in their organs is greatly species-specific. In addition, many factors can influence metal uptake by fish, including sex, age, size, reproductive cycle, swimming pattern, feeding behavior, and geographical location [27,28]. Heavy metals are difficult to absorb from the gastrointestinal tract in the form of free ions, but their organic compounds are absorbed almost completely [29]. Methylmercury can accumulate in fish and other species, particularly in long-lived and larger predators [30]. It has been found that nearly all mercury in wild fish is MeHg [31], which is almost entirely accumulated via dietary uptake [29]. For this reason, it is necessary to monitor both the aquatic environment and the fish living there to identify variable abiotic and biotic factors that can increase metal levels in fish organs [32].

The gills and liver are key organs in fish metabolism and toxicopathology, and hence have traditionally been analyzed in monitoring environmental pollution and fish health [33,34,35]. The liver plays a central role in the binding, storage and redistribution of mercurals which enter peripheral circulation [36]. Fishery products play an important role in human nutrition, being components of healthy diets and a traditional cultural asset; therefore, muscle tissue is analyzed to assess the safety aspects of fish consumption by humans [34,35,37,38]. Even low levels of MeHg may interfere with the complex developmental processes in the fetal brain, which include cell differentiation, migration and synaptogenesis [39]. As recommended by the US EPA [40], to maximize the benefits of eating fish while minimizing exposure to mercury, consumers should mainly eat types of fish which are low in mercury and limit consumption of fish which typically have higher levels. 

In the present study, the level of MeHg in fish was calculated based on total mercury (THg) according to the EFSA recommendations [41,42]. Since mercury is ubiquitous in the environment, humans, plants and animals are constantly exposed to it [43]. Therefore, the aim of this study was to determine the mercury contamination of abiotic elements, i.e., water and sediment, and biotic elements, i.e., the common reed (*Phragmites australis*), pike (*Esox lucius*), bream (*Abramis brama*) and roach (*Rutilus rutilus*), of two trophically diverse lakes in north-western Poland. The study also examines the following: (1) the relationship between Hg content in water and sediments and its distribution in the organs of common reed and fishes; (2) the impact of selected water and sediment parameters (transparency, temperature, pH, O_2_) and other factors (season, sex, body weight, total length) on mercury accumulation in biota; (3) whether the process of mercury biomagnification occurs between the examined fish representing different trophic levels (pike, predator; bream, bentophagus; roach, herbivore). Moreover, due to the popularity of the tested fish species among Polish consumers, the dietary uptake of Hg (converted to MeHg) was also assessed based on estimated daily intake (EDI), tolerable weekly intake (EWI) and target hazard quotient (THQ).

## 2. Materials and Methods

### 2.1. Description of the Study Area

Lake Ińsko (53°26′36″ N, 15°32′33″ E) and Wisola (53°24′46″ N, 15°32′49″ E) are located in north-western Poland. Compared with other lakes of the Polish Plain, Lake Ińsko is large (486.6 ha) and deep (maximum depth 41.3 m, mean depth 12.9 m) [44]. Lake Wisola has a water surface area of 181.5 ha, a mean depth of 5.9 m and a maximum depth of 15.4 m [31]. The two lakes differ in the degree of eutrophication and anthropopressure. Studies conducted in the years 1970–2010 showed significant changes in the water quality of Lake Ińsko and its trophic gradient, from mesotrophy to signs of eutrophication [44]. Based on the saturation of the hypolimnion with oxygen, Filipiak et al. [31] classified Lake Ińsko as a α-mesotrophic subtype (oxygen saturation > 20%) and Lake Wisola as β-mesotrophic (oxygen saturation < 20%).

Lake Ińsko was negatively influenced by nearby urban development (the town of Ińsko), which was reflected in a higher concentration of nitrogen, phosphorus compounds and sulphates [45]. However, at the end of the 1990s and in the 2000s, the lake reverted to a mesotrophic state [44] characterized by a moderate susceptibility to deterioration and the second class of water purity [46]. According to Kubiak et al. [44], the improvement most likely resulted from the changes in soil use in the catchment area, as less phosphorus and nitrogen compounds flowed into the lake, as well as from the improved sewage disposal system in the town of Ińsko. Lake Wisola is a flow-through reservoir fed by the Iński Canal (W7) (Figure 1); however, the outflow sometimes stops as a result of periodic stoppages in the inflow of water through the Iński Canal, and Lake Wisola becomes drainless.

### 2.2. Materials and Field Methods

Research material was collected from both lakes eight times (twice in every major season of the year), with the tested material consisting of various abiotic and biotic components, including water, bottom sediments, common reed (*Phragmites communis* Trin.) and three fish species representing different trophic levels: carnivorous pike (*Esox lucius* L.); benthivorous bream (*Abramis brama* L.) and planktivorous roach (*Rutilus rutilus* L.). Water samples from surface and near-bottom (approx. 0.5 m from the bottom) layers were collected in polyethylene containers with a capacity of 2.5 L after they had been rinsed several times with water from the lake. A range of water quality data was obtained directly (in situ) from field measurements: temperature (°C), with an accuracy of 0.5 °C, dissolved oxygen (oxygen meter, Elmetron, Zabrze, Poland), acidity (pH) with an accuracy of ±0.1 pH (pH meter, pHScan BNC, Singapore, Republic of Singapore) and transparency (Secchi disk, Aqualabo, Champigny sur Marne, France). Bottom sediments were collected using a van Veen dredge (KC Denmark A/S, Silkeborg, Denmark) and, after mixing, packed in polyethylene ziplock bags and placed in a portable refrigerator.

Reeds were collected from the strip of emergent vegetation forming rushes at the coastal points marked in yellow in Figure 1. The plants were washed on site with lake water, drained on blotting paper, divided into four parts, i.e., leaves, stems, rhizomes and roots, and placed in polyethylene bags.

The fish (pike, bream, roach) were obtained from catches made in both lakes by a fish farm based in Ińsko. Equal numbers of male (m) and female (f) fish were taken for the study: a total of 480 fish specimens. The respective ranges (min–max) of the weight and total length of the fish caught in Ińsko and Wisola lakes were as follows: pike (408–1260 g; 40.0–56.0 cm) and (263–1480 g; 35.0–64.0 cm), bream (300–1560 g; 30.0–51.0 cm) and (270–1400 g; 30.0–74.6 cm), roach (38–243 g; 15.0–28.0 cm) and (48–232 g; 17.3–27.5 cm). 

### 2.3. Material Pre-Treatment

In the laboratory, mercury analyses were carried out on an ongoing basis in the water, which was stored in refrigerators (4 °C) until the analyses. Parts of the bottom sediments were used to determine pH (pHscan BNC pH meter) and organic matter. The remaining material was frozen in Petri dishes, after which it was lyophilized in the Heto LyoLab 3000 apparatus (Thermo Fisher Scientific, Allerød, Denmark). The lyophilized bottom sediments were ground in an agate mortar, sieved through a 350 μm nylon sieve and stored in sealed polyethylene containers until analysis. The isolated common reed organs were cut into small particles using stainless steel tools, put in polyethylene bags and placed in the freezer (−20 °C). The fish were first measured and weighed and then frozen in their entirety. After partial thawing, the following organs were taken out for further analysis: dorsal muscles, skin, gonads, kidneys, livers, spleens, gill lobes, digestive tracts (these were divided into stomach and intestine in pike and bream) and food content. The collected material was placed in polyethylene bags and stored at −20 °C until the analysis.

Dry matter content was calculated based on the moisture content in the fish organs and tissues (%). For this purpose, the moisture content in the tested material was analyzed in accordance with the AOAC standard method 950.46 [47]. An analysis of organic matter content in sediments was carried out using the Loss on Ignition method [48]. The bottom sediments were divided into two groups based on the percentage share of organic matter content: organic (>5%) and mineral (<5%).

### 2.4. Samples Preparation

Merck’s highest purity reagents (Merck KGaA, Darmstadt, Germany) and deionized water from the Barnstead EASYpure UV instrument (Thermo Fisher Scientific, Allerød, Denmark) were used for all laboratory procedures. The solid material was weighed on a WPS 360/C analytical balance (RADWAG, Radom, Poland) with an accuracy of 0.001 g. Samples of water, bottom sediments and plants were prepared as three parallel replicates. From the fish tissues and organs, every tenth sample was analyzed three times. In parallel, blank samples and reference materials were prepared.

#### 2.4.1. Water Samples

For mercury determination, 1 L of water samples was poured into quartz glass beakers. Then, 20 mL of KMnO_4_ and 15 mL of concentrated H_2_SO_4_ were added and the samples were placed in an incubator at 60 °C for one hour. After cooling the samples, the excess KMnO_4_ was reduced by the addition of 50% hydroxylamine hydrochloride solution (HONH_2_·HCl). The samples were transferred to glass separatory funnels and Hg was extracted twice, using 10 mL of chloroform dithizone solution each time. The extracts were collected in another separatory funnel containing 50 mL of 0.25 N H_2_SO_4_. Next, 10 mL of 40% KBr solution was added and the samples were shaken for one minute to release mercury from the dithizone layer. The dithizone layer was discarded and the samples were adjusted to pH 6 using a pH meter. The samples were quantitatively transferred to volumetric flasks with a capacity of 100 mL and made up to the mark with deionized water. Deionized water was tested in parallel with lake water samples as a reagent blank.

#### 2.4.2. Common Reed, Sediment and Fish Samples

The following materials were weighed for the analysis, with an accuracy of ±0.010 g: bottom sediments (0.25 g dw), common reed (2 g dw), fish muscles (5 g ww) and other organs (1 g ww). The materials were mineralized in a mixture of concentrated HNO_3_ and HClO_4_ at the ratio of 4:1. Fish tissues and organs were mineralized with 5 mL of the mixture and 15 mL were used for sediments and common reeds. Fish and bottom sediment samples were left at room temperature for 24 h, while plant samples in general for 48–72 h. After that, the samples were placed in an incubator at 70 °C for 24 h (plant samples usually 3 times longer). Opened samples were left in a water bath for 2 h to remove nitrogen oxides, filtered into 100 mL volumetric flasks and filled up with deionized water.

### 2.5. Mercury Determination

Total mercury content was determined by atomic absorption spectrometry using the cold vapor technique (CVAAS) in a Bacharach Coleman MAS-50 mercury analyzer (Bacharach Inc., Pittsburgh, PA, USA). Mercury quantification was performed on the basis of an external standard curve. A standard curve was prepared using a certified standard mercury solution (1000 mg/L, Merck, Germany). Mercury content was determined at a wavelength of λ = 253.7 nm. Calibration was performed using five Hg concentrations in the range of 0.20–10.0 µg/L. For six independent calibrations, an average linear determination coefficient (R2) of 0.9986 was calculated. The limit of detection (LOD) was 0.05 µg/L and was based on the 3 SD criterion (LOD = 3 SD B/m, where m means the slope of the calibration curve and SD B means the standard deviation of 10 successive measurements of blank samples (B)).

After calibration of the mercury analyzer, the prepared samples (100 mL), were transferred to a reaction vessel where mercury ions were reduced to the metallic form with 5 mL of 10% SnCl_2_ and the concentration of mercury vapors was subsequently determined. Blank samples, i.e., samples devoid of the test material, were also analyzed in parallel with the test material. The Hg concentrations are given in the following units, depending on the tested material: water, ng/L, bottom sediments and reed, mg/kg dw, and fish, mg/kg ww.

### 2.6. Analytical Quality Control

The accuracy and precision of mercury determination was confirmed using the standard addition method for water samples, and by appropriate certified reference materials in other samples: for fish—DOLT-2 (dogfish liver, National Research Council of Canada (NRC-CNRC)) and Fish-Paste-2 (Quality Control Reference Material LGCQ1005); for the common reed—INCT-MPH-2 (Mixed Polish Herbs, Institute of Nuclear Chemistry and Technology, Warsaw, Poland); and for sediments—MESS-3 (Marine Sediment, National Research Council of Canada (NRC-CNRC)). The results obtained for total Hg in the reference materials compared with the certified values, as well as relative standard deviation (RSD) are presented in Table 1.

### 2.7. Accumulation and Bioaccumulation Factors of Mercury

#### 2.7.1. Mercury Accumulation in Bottom Sediments

Accumulation factors (*AFs*) for Hg in bottom sediments were calculated for each site (Figure 1) separately, according to Formula (1):(1)$$AF={\frac{C_{s}\times 10}{C_{w}}}^{6}$$
where:

*AF*, Hg accumulation factor for bottom sediments;

*C_s_*, Hg content in bottom sediments (mg/kg dw);

*C_w_*, mean concentration of Hg in surface and bottom water (ng/L).

#### 2.7.2. Mercury Bioaccumulation in the Organs of Common Reed and Fish

Bioaccumulation factors (*BAF*s) for Hg in common reed organs were calculated for each site (Figure 1) separately, according to Formulas (2) and (3) [49]. For fish, the average content in the season and the total average content for a given species and lake were taken into account:(2)$$BAF_{w}={\frac{C_{o}\times 10}{C_{w}}}^{6}$$
(3)$$BAF_{s}=\frac{C_{o}}{C_{s}}$$
where:

*BAF*, Hg bioaccumulation factor in the organs of common reed/fish;

*C_o_*, Hg content in the organs of common reed (mg/kg dw)/fish (mg/kg ww);

*C_w_*, mean concentration of Hg in surface and bottom water (ng/L);

*C_s_*, mean content of Hg in bottom sediment (mg/kg dw).

*BAF* reflects the uptake of a chemical from water, from contaminated food and from sediment consumed by the organism. As bioaccumulation includes chemical exposure through diet, it also reflects the potential of biomagnification [49,50].

### 2.8. Assessment of Hg Dietary Intake

The results for total Hg content in fish muscles obtained in this study were compared with the highest permissible levels in fish and fish products set in the Commission Regulation (EC) No. 1881/2006, as amended [51]. For risk assessment, the total mercury values obtained in this study were converted to methylmercury based on the two assumptions proposed by the European Food Safety Authority [41,42]: (1) 100% of the total mercury measured in fish meat is methylmercury [41]; (2) the total mercury content in fish is converted to methylmercury and inorganic mercury by applying conversion factors of 1.0 for methylmercury and 0.2 for inorganic mercury [42].

In order to assess the possible health effects resulting from the intake of Hg from the muscles of the studied fish, the following were calculated: estimated daily intake (EDI) (µg/kg body weight (BW)) and target risk factor (THQ), as set by the US Environmental Protection Agency [52,53]. The above indicators assess the potential non-carcinogenic risk to human health. When THQ > 1, systemic effects may occur. If the EDI is equal to or greater than the reference dose (RfD), the exposed human population will experience a health hazard. Moreover, the estimated weekly intakes (EWI) (µg/kg bw) were calculated by multiplying the EDI by seven, i.e., the number of days in a week. The obtained EWI values were compared with the tolerable weekly intake (TWI) of 1.3 μg/kg/week set by EFSA [41] on the basis of the prenatal neurodevelopmental toxicity for MeHg. The number of servings of fish per week a given population group would need to eat to reach the TWI for methylmercury was also calculated.

The EDI and THQ were calculated using the Equations (4) and (5) below. In the calculation, the following assumptions were made: Hg is the mean content of the calculated MeHg in the muscles of pike, bream and roach; adult body weight is 70 kg; fish consumption is 300 g (two servings of 150 g per week) [54].
(4)$$\mathrm{EDI}=\frac{C\times \mathrm{IR}}{\mathrm{BW}}\;$$
(5)$$\mathrm{THQ}=\frac{\mathrm{EF}\times \mathrm{ED}\times \mathrm{IR}\times C}{\mathrm{RfD}\times \mathrm{BW}\times \mathrm{AT}}\times 10^{-3}$$
where: C, Hg content in fish (mg/kg ww); IR, fish ingestion rate (g/day); BW, body weight (kg); EF, exposure frequency (365 days/year); ED, exposure duration (70 years); RfD, oral reference dose (mg/kg/day) for Hg (1.0 × 10^−4^); AT, averaging exposure time for non-carcinogens (365 days/year × ED).

### 2.9. Statistical Analysis

All statistical calculations were performed using Statistica ver. 8.0 software (Statsoft Inc., Tulsa, Oklahoma, OK, USA). The statistically significant differences in Hg content between the organs of common reed and between fish species were calculated by means of the post hoc Duncan’s Honestly Significant Difference test at a significance of *p* ≤ 0.05. Correlation coefficients were also calculated to assess the relationship between the physicochemical parameters of water and sediment, as well as between the environmental parameters (temperature, pH, O_2_, transparency, Hg content) and Hg accumulation in the reed and fish organs.

## 3. Results

### 3.1. Mercury Concentration and Other Parameters

#### 3.1.1. Abiotic Elements (Water and Bottom Sediments)

Due to its very low concentration in water, mercury was expressed as ng/L. In the surface and bottom waters of both lakes and in their outflows, mercury concentrations did not differ significantly throughout the study period and typically ranged from 0.9 to 1.5 ng/L (Table 2). Higher concentrations exceeding 4 ng/L were found in the water collected in autumn and summer from the tributary of Lake Wisola at site W7 (the Iński Channel) and in autumn at the outflow site W6 (Figure 1). The pH value of the waters of the studied lakes in the entire study period ranged from 7.1 to 9.2 in the case of Lake Ińsko and from 7.1 to 9.4 for Lake Wisola. Oxygen content varied depending on the season and water layer: from 3.6–12.6 mg/L in Lake Ińsko to 0.2–12.5 mg/L in Lake Wisola; the latter range was wider due to periodic oxygen deficiency in the bottom layer (Table 2). The tributary of Lake Ińsko was characterized by a pH of 6.9–7.6 and an oxygenation of 6.1–10.0 mg/L and of Lake Wisola by pH 6.8 to 7.9 and an oxygen content from 2.0 to 10.4 mg/L. The waters of Lake Ińsko were characterized by very good transparency (4.0 to 6.0 m) throughout the research period, while transparency decreased periodically to 1.5 m in Lake Wisola, with an average of 2.6 m.

The average mercury levels in organic sediments (organic matter > 5%) amounted to 0.109 mg/kg dw in Lake Ińsko and 0.148 mg/kg dw in Lake Wisola, and were significantly higher compared with the group of mineral sediments (organic matter < 5%), where the average mercury levels were 0.014 and 0.019 mg/kg dw, respectively (Table 3). The average Hg content over the entire study period was higher in the organic sediments from Lake Wisola than in those from Lake Ińsko. In the mineral sediments, no significant differences in mean mercury content were found between individual seasons for either lake; however, in the organic sediments, mercury content varied depending on the season for both lakes (Figure 2). In Lake Ińsko, the Hg level in the sediments was highest in summer, followed by autumn > spring > winter, with a significant difference only between summer and winter. In the sediments of Lake Wisola, Hg content in all seasons was significantly higher than in winter. During the research period, organic bottom sediments were characterized by a pH of 6.5–7.2 in Lake Ińsko and 6.4–7.7 in Lake Wisola, while the pH values for mineral sediments were 6.2–7.5 and 6.2–7.5, respectively. The share of organic matter in the sediments depended on the sampling site and ranged as wide as 0.8–28.1% in Lake Ińsko and 0.9–47.9% in Lake Wisola. With regard to Hg concentrations in the water of the two lakes, accumulation factors in the organic sediments ranged from 48,000 to 292,000 in Lake Ińsko and from 76,500 to 196,000 in Lake Wisola. In the mineral sediments, mercury accumulation factors were 3500–35,500 in Lake Ińsko and 2000–89,000 in Lake Wisola (Table 3).

#### 3.1.2. Biotic Elements: Common Reed, Fish (Pike, Bream, Roach)

The average Hg content in common reed from lakes Ińsko and Wisola ranged between 0.006 and 0.017 mg/kg dw (Table 4). In general, Hg accumulation was highest in the root, followed by leaves > stems > rhizome. However, in the case of plants from Lake Ińsko, a significant difference was only confirmed between Hg content in rhizomes and in other organs (*p* < 0.05). More significant differences (*p* < 0.05) were confirmed in the reed from Lake Wisola (Table 4). In both lakes, the accumulation of mercury in reed organs differed depending on the season and, in general, plants harvested in spring and summer contained less mercury than those collected in autumn and winter. The mean content of Hg in leaves was within the ranges of 0.003–0.014 (Lake Ińsko) and 0.005–0.020 (Lake Wisola) (mg/kg dw). Leaves collected in autumn contained significantly more Hg than those collected in spring and summer. In winter, leaves were not analyzed due to their residual occurrence. The remaining organs collected in autumn and winter contained significantly more mercury than those from spring and summer; the mean Hg content had the ranges, respectively, for Lake Ińsko and Lake Wisola, 0.002–0.030 and 0.016 in stems, 0.002–0.012 and 0.002–0.021 in rhizomes and 0.004–0.034 and 0.009–0.027 in roots (mg/kg dw).

The mercury content in the organs and tissues of the examined fish throughout the study period are presented in Table 5. The Hg level ranged from <0.001 to 0.460 mg/kg ww in pike, <0.001 to 0.090 mg/kg ww in bream and 0.001 to 0.155 mg/kg ww in roach. Mercury detected in the food content of all of the studied fish ranged from 0.025 to 0.125 mg/kg ww and was the lowest in the case of bream (Table 5).

The tested organs accumulated as much as 40.7 times more Hg than the muscles. In general, the kidneys and livers of the pikes and breams play a major role in mercury accumulation, but in the case of roach, the muscles turned out to be more important. Differences related to the sex of the studied species were statistically confirmed only in cases of the spleen of pike from Lake Ińsko (r = 0.78) and gills of pike from Lake Wisola (r = 0.89). In both cases, higher levels were found in the organs of males. No sex-related differences in Hg accumulation were found in the organs of the roaches or breams. Although the fish from both lakes demonstrated similar distributions of mercury bioaccumulation, some intra-species and inter-species differences were found in mercury content (Duncan’s test, *p* < 0.05). For instance, Hg level was found to be significantly higher in the muscles and gills of pikes, muscles of breams and spleens of roaches from Lake Wisola than in those of the same fish species from Lake Ińsko. By contrast, more mercury was detected in the kidneys, digestive tract and ingesta of breams and in the skin of roaches from Lake Ińsko than in those of breams and roaches from Lake Wisola (Table 5). Statistically significant inter-species differences were found with regards to fish muscles. The level of mercury in the muscles of roach, from both lakes Ińsko and Wisola, was significantly higher than in the muscles of pikes and breams.

The correlation coefficients between the mercury content of the gills and the water (r = 0.50–0.98) indicate that the respiratory tract played a significant role in Hg uptake. In some cases, oral intake also played an important role, as indicated by the high correlation coefficients between mercury content in the livers and digestive tracts of roaches (r = 0.69) and in the livers and digestive tracts of breams from Lake Ińsko (r = 0.74). In addition, the high bioaccumulation factors (*BAF*) calculated for the digestive tracts of all the fish species from both lakes indicate that the digestive system plays a role in mercury uptake (Table 6). Additionally, the *BAFs* show that pike also took up mercury to a large extent through the gills, while the skin was of marginal importance. In breams, bioaccumulation factors indicate Hg uptake mainly through the digestive system and to a much lesser extent through the skin and gills. In roach, the gastrointestinal tract and skin were of comparable importance in mercury uptake, while the gills played a minor role (Table 6).

Statistical analysis showed no effect of body size (mass, length) on mercury accumulation in tissues and organs of pikes. However, a significant positive effect of both weight and length on the accumulation of mercury in the muscles and spleens of breams from both lakes and in the spleens of roaches from Lake Ińsko was found (Table 7). In addition, a positive correlation was found between the length of bream from Lake Ińsko and the level of Hg in the gonads, and the weight of bream from Lake Wisola and its accumulation in the stomach. Negative correlations were observed in roach kidneys from Lake Wisola.

The potential human health risk assessment for adults is presented in Table 8. Assuming a 100% share of methyl mercury in fish [41], the estimated daily intake (EDI) was in the range of 0.001–0.131 µg/kg body weight (bw), with the average value not exceeding 0.041 µg/kg bw, obtained for the muscles of the roach from Lake Wisola. Assuming a lower share of methylmercury in its total content [42], the results were correspondingly lower. The mean estimated weekly intake (EWI) for total mercury (µg/kg bw/week) resulting from the consumption of fish muscles from lakes Ińsko and Wisola, respectively, was pike (0.034; 0.146), bream (0.021; 0.056), roach (0.219; 0.287). The obtained mean EWI values ranged from 0.024% (bream from Lake Ińsko) to 0.240% (roach from Lake Wisola) of the Tolerable Weekly Intake (TWI). The target hazard quotient (THQ) for the average mercury content of all fish species tested was well below 1, i.e., the threshold value.

## 4. Discussion

Throughout the study period, the waters of Ińsko and Wisola lakes and their tributaries and outflows, contained trace amounts of mercury (0.40 to 5.4 ng/L) (Table 2); these complied with the limits for uncontaminated rivers and streams, i.e., 1–7 ng/L [55]. The recorded values were also low compared to other criteria for unpolluted waters, i.e., 70 ng/L [56] or 100 ng/L [57,58], as well as for the water background (10 ng/L) [59]. In addition, the tested waters met the criteria for assessing the quality of water intended for human consumption with regard to Hg concentration [60]. The low concentrations of Hg in the waters of lakes Ińsko and Wisola resulted both from its very low solubility in water [61] and its low accumulation in bottom sediments. 

The mercury content in the organic sediments ranged from 0.034 to 0.352 mg/kg dw (mean 0.109 mg/kg dw) in Lake Ińsko and from 0.053 to 0.244 mg/kg dw (mean 0.148 mg/kg dw) in Lake Wisola (Table 3). Similar sediment Hg contents (0.014–0.236 mg/kg dw) were reported by Sychra et al. [62] in fish ponds in two regions of the Czech Republic. According to the Canadian Sediment Quality Guidelines for the Protection of Aquatic Life [63], the Hg values in organic sediments are near or higher than the threshold effect level (TEL) (0.13 mg/kg dw) and the probable effect level (PEL) (0.7 mg/kg dw). Hence, in organic sediments, Hg accumulation is generally at a level at which adverse effects may occur occasionally or even frequently. The organic group of sediments was also characterized by an increased Hg accumulation compared with the background level determined for bottom sediments of lakes in Poland (0.05 mg/kg dw) [58]. However, only the maximum concentrations recorded in the sediments of Lake Wisola exceeded other background values (mg/kg dw): >0.1 [64], 0.2 [65] and 0.3 [59], and only slightly. The mercury content in the examined sediments was also well below the threshold of 1 mg/kg dw indicating contamination of spoil from the dredging of sea basins, watercourses and other water reservoirs [66].

In both lakes, organic matter content appeared to have a significant impact on mercury accumulation in sediments, as confirmed by positive linear correlation coefficients (r) ranging between 0.33–0.78. Similar observations with regard to mercury and other trace elements have often been reported in literature [11,67]. High Hg accumulation factors (Table 3), especially in the case of organic bottom sediments, seem to confirm that organic matter plays an important role in binding Hg in the studied lakes. Moreover, good water parameters for Lake Ińsko, especially oxygen saturation, pH and transparency, may have contributed to a stronger binding of Hg ions in the sediments. Many studies have suggested that organic matter substantially reduces the bioavailability and risks of mercury in soils and sediments [68]. Some authors also point out that the abundance of reduced sulfur sites on organic matter molecules strongly bind to Hg, resulting in its immobilization and reduced bioavailability [69,70]. This could explain why the concentrations of mercury in the examined hydrobionts were low despite the fact that in some cases, sediment Hg content exceeded natural background levels [58]. On the other hand, it has also been reported that organic matter greatly accelerates Hg methylation and increases the risks of Hg exposure [68]. Accumulation of mercury in organic sediments of both lakes dependent to some extent on the season. In Lake Ińsko, the lowest Hg content was found in winter, but a significant difference was only observed in spring. A higher content of Hg in the organic sediments of Lake Wisola was found in warmer seasons, when plankton blooms occur in this lake, and the lowest was observed in winter (Figure 2). In addition, this lake also demonstrated a decrease in oxygen concentration, an increase in pH and low visibility of the Secchi disk (average 2 m) during the warmer seasons, especially in spring.

Similar observations were noted by Bełdowski [69], who attributed the increased concentration of total mercury in sediments during plankton blooms to the supply of mercury assimilated by phytoplankton from the water column. In addition, similarly to the presented research, this study recorded a lower mercury content in the sediments collected in winter, which was explained by the possibility of heavy icing cutting off the surface runoff from the area.

Garcia-Ordiales et al. [70] propose that total mercury (THg) is more likely to bind in the matrix of sediment in winter and be more mobile in summer, when it is released to surface water or leached into surface runoff. Monteiro et al. [71] attribute the presence of higher concentrations of mercury in the methyl form in summer than in winter to the process of “seasonal” methylation, which occurs in both the uppermost and the deeper sediment layers. However, studies of sediments from the contaminated sites of the Hyeongsan River estuary in South Korea [72] found the highest total mercury (THg) and methylmercury (MeHg) levels in the autumn, and found that these remained at a similar level in the winter, before decreasing in the summer. In addition, in contrast to whole sediment, pore water demonstrated higher mercury levels in summer and these gradually declined during autumn and winter.

The mercury content of the reed samples from Lake Ińsko and Lake Wisola ranged from 0.001 to 0.099 mg/kg dw; these values were lower than those recorded in reed from Lake Świdwie (Poland) [73] and were below those considered excessive (toxic) in the above-ground parts of plants (1–3 µg/g dw) [65]. The obtained results were also within the background range for Hg in edible plants (0.003 to 0.09 mg/kg dw) [55]. The range of mercury content in reed stems (0.001–0.045 mg/kg dw) was similar to those reported in studies of fish ponds in two regions of the Czech Republic [62]. Much higher mercury levels were found in the macrophyte *Eleocharis elegans* growing in aquatic systems of abandoned gold mines in Western Colombia: the level was 0.16 µg/g dw in underground biomass and 0.05 µg/g dw in the aerial biomass [7].

Another study tested total mercury (THg) levels in 259 wild plants of 49 species from 29 families that grew in the heavily mercury-contaminated cinnabar (calcin) wasteland of southwestern China. Of all the species studied, four were characterized by significantly elevated THg concentrations in the shoots and/or roots, reaching 100 μg/g, while three demonstrated content not exceeding 0.5 μg/g [15]. In the present study, in both lakes, the highest level was found in the roots, followed by the leaves > stems > rhizomes. According to Kabata-Pendias [55], the uptake of Hg in plants is mainly through roots, where it is accumulated; its translocation to the shoots is relatively small.

For the common reeds, the mean bioconcentration coefficient of mercury, i.e., calculated in relation to its share in sediments, was close to 1, e.g., in reed leaves and roots from Lake Ińsko, and in some cases, even exceeded 1, e.g., in reed roots from Lake Wisola (Table 4). According to Quian et al. [15] exceeding this value suggests that the plant is able to extract mercury from the soil. The same authors observed high concentrations of both THg and Mg in the fern (*Eremochloa ciliaris*) and also reported BCF values above 1. They concluded that *E. ciliaris* might be a suitable candidate for phytoextraction. Higher bioaccumulation rates of mercury in roots compared to the rest of the reed plant indicate that they are a good indicator of changes in Hg content in the environment [74].

Moreover, in the case of Lake Ińsko, a positive correlation (*p* < 0.05) was found between mercury content in roots and its share in bottom sediments (r = 0.48). In Lake Wisola, Hg in sediments was positively correlated with its content in all parts of the reed (r = 0.42–0.70), and the concentrations in roots and leaves correlated with the concentrations in water (r = 0.46–0.50). Hence, it can be concluded that the reed is sensitive to changes in mercury content in the surrounding environment.

Regarding the fish from lakes Ińsko and Wisola it was found that, generally, roaches and breams contained much less mercury in organs and food content than pikes (Table 5). However, the exceptions were roach muscles and skin. In this study, the following patterns of mercury bioaccumulation were observed in the fish from both lakes (in decreasing order of Hg content): pike: kidneys > liver > intestines > stomach > spleen > gonads > skin; bream: liver > kidneys > spleen > digestive tract > skin > gills; roaches: muscles > liver > kidneys > digestive tract > gills > gonads. Other authors have also found a higher mercury content in the liver than in the muscles of pikes [34]. The present results are also in line with those reported by Cuvin-Avalar and Furness [75], who found that while the kidneys are the target organs, the gills, liver and gastrointestinal tract may also contain significant amounts of mercury. 

The Hg accumulation pattern in roaches from both lakes is compatible with general observations that the highest levels of Hg are present in fish muscles and the lowest in gonads [55]. Similar observations have been noted in other studies [73]. The importance of muscles in mercury bioaccumulation has already been pointed out by Håkanson [76], who recorded its higher Hg level in muscles than in kidneys and livers. Our present finding indicate that mercury was accumulated to the greatest extent through the liver, kidneys and gastrointestinal tract, and to a lesser extent through the skin and gonads. Significant differences related to the sex of the studied species were only recorded in the case of the spleen of pike from Lake Ińsko (r = 0.78) and gills of pike from Lake Wisola (r = 0.89). In both cases, higher levels were found in the organs of males. No sex-related differences in Hg accumulation were found in the organs of roaches or breams. In another study of fish of various trophic levels collected along the Maderabadany River, no clear pattern of differences was found between males and females in 90% of 41 studied species [77].

In the present study, few statistically significant relationships were found regarding the influence of the body size on mercury bioaccumulation (Table 7). Similarly, no significant correlations were reported between the mercury content in the muscles of six fish species (including roach) from Skadar Lake (Montenegro) and their size [78]. In our research, a positive correlation between mercury accumulation and weight and length was found in the case of bream from Lake Ińsko.

Regarding mercury levels in fish muscles, in both lakes, the mean Hg content in roach muscles (0.051 and 0.067 mg/kg) was significantly higher than in pikes (0.008 and 0.034 mg/kg) and breams (0.005 and 0.013 mg/kg). Our results correspond partly to those reported by Sonesten [25] for roaches from 78 lakes in Sweden (0.04–0.31 mg/kg ww) and those reported by Chałabis-Mazurek et al. [79] for roach (0.0123–0.0499 mg/kg) and pike (0.0185–0.0255 mg/kg) from three lakes in Poland. Similar mercury levels were also reported in pike muscles from Lake Świdwie [73] and in fish from West Pomerania (mean 0.015–0.030 mg/kg ww) [80]. In contrast, a higher Hg content was recorded in pike from the Moulouya River in the eastern region of Morocco [38], in the muscles of Croatian fish [35] and fish from the Elbe River [81].

In the light of the Environmental Quality Standard (EQS) for mercury and its compounds in fish (0.02 µg/kg bw) [82] as well as the data on the natural background level of Hg in fish muscles (0.02–0.150 mg/kg bw) collected by Kozak et al. [83], all examined fish from both lakes demonstrated low mercury content int their muscles (mean: 0.005–0.067 mg/kg bw). However, in most other organs of pike and roach, the mean Hg content (0.003–0.086 mg/kg ww) exceeded the EQS limit (Table 5).

Considering only the mercury content in the muscles, our findings indicate that the relationship between herbivorous roaches and predatory pikes is not subject to biomagnification; however, this process may be possible between breams and pikes. Our findings are consistent with those of by Chałabis-Mazurek et al. [79], who report a higher mercury content in the muscles of planktonophages and benthophages than in predators. However, in the natural environment, the predator feeds on the whole prey. Therefore, in order to more accurately determine the food relationships in the food chain, it would be necessary to take into account many other variables affecting the bioaccumulation of mercury in the whole fish. Such research would have to compare a range of variables characterizing the studied species, including age, fish size, growth rate, behavior, ontogenesis (including sexual maturity), as well as the sampling season and the rate of metabolism and food intake [21].

However, our findings indicate significantly higher levels of mercury in most organs other than muscles in pike compared to bream and roach, which seems to support the occurrence of biomagnification between the studied species. Even so, it would be important to take into account the weight of individual organs and tissues and their share in the total weight of the fish in future studies.

### Health Risk Assessment for Fish Consumers

The mercury levels recorded in fish muscles were within the maximum permissible levels applicable in the European Union, i.e., 0.5 mg/kg for roach and bream and 1.0 mg/kg for pike [51]. The average estimated weekly intake (EWI) for total mercury (µg/kg bw/week) calculated in respect of the consumption of fish muscles from Ińsko and Wisola lakes, respectively, was as follows: pike (0.034; 0.146), bream (0.021; 0.056), roach (0.219; 0.287). On average, the calculated EWI values ranged from 0.02% to 0.32% of the tolerable weekly intake (TWI), the lowest value being for breams from Lake Ińsko and the highest one for roaches from Lake Wisola (Table 8). The above calculations were made for the recommended weekly consumption of fish, assuming two servings of 150 g per week [41], which is overestimated in relation to the Polish consumer, who on average consumes only 70 g of fish and seafood per week [84]. Even taking into account the maximum determined mercury content obtained for pikes from Lake Wisola (0.214 mg/kg), an adult (70 kg bm) would have to consume as much as 30 kg of pike to achieve 100% TWI for methylmercury. Other authors noted higher levels of mercury in fish muscles, which resulted in a greater coverage of TWI, ranging from 5.80 to 21.3% depending on the age group and socio-economic status of the consumer [85]. Pike was reported by Latvian researchers as one of the main fish contributors affecting the supply of MeHg with the diet, with the mean Hg content in pike muscles being 0.394 mg/kg [41]; this significantly exceeds the average and even the maximum values obtained in this study (Table 5).

The estimation of non-carcinogenic health risks based on the target hazard quotient (THQ) did not reveal any obvious risks for fish consumers (THQ < 1). The average THQ values for all of the examined fish were in the range of 0.03–0.4, with roach muscles as the major risk contribution (Table 8). Similar results were reported by Mahjoub [38] for pike and other fish species. However, taking into account the maximum levels of Hg found in muscles, the THQ threshold value was exceeded in the case of pike from Lake Wisola (1.31 for THg and 1.09 for MeHg) and was close to 1 for THg in the muscles of roach from Lake Wisola (0.95) (Table 8).

## 5. Conclusions

Despite some fluctuation, mercury concentrations in the waters of lakes Ińsko and Wisola and their tributaries and outflows were generally within the range characteristic of unpolluted waters. The bottom sediments of Lake Wisola had a higher Hg content than the sediments of Lake Ińsko. In both lakes, the mercury level in the sediments was positively correlated with the share of organic matter, which probably contributed to the immobilization of mercury and its reduced bioavailability; this seems to be confirmed by the low mercury content in the tested organisms.

The common reed absorbed Hg mainly from water and showed the highest level of bioaccumulation in roots. The higher bioaccumulation rates in roots compared to leaves, stems and rhizome suggest that the plant could be a good indicator of mercury changes in the environment.

Among the fish from both lakes, a cross-species comparison showed that pikes had the highest levels of mercury in most of the organs tested. Roaches, on the other hand, accumulated most Hg in the muscles and skin, indicating that there was no biomagnification in the planktivorous-predatory and benthic-predatory fish systems. The target organs of mercury accumulation in pike and bream were the kidneys and liver, while the muscles were more important in the case of roach.

The toxicological assessment found the muscles of the studied fish did not pose a threat to consumer health in terms of mercury content. Moreover, neither the calculated THQ values nor the recorded Hg intake, with regard to TWI, raise any health concerns for adult consumers of the examined fish muscles.

Our results indicate the need for more detailed studies to determine the nutritional relationships between pike (predator) and roach (herbivore) in order to verify the process of biomagnification in the studied lakes. The presented results can be used as a source of data on mercury levels in lakes not subject to increased anthropopressure.

## Figures and Tables

**Figure 1 animals-13-00697-f001:**
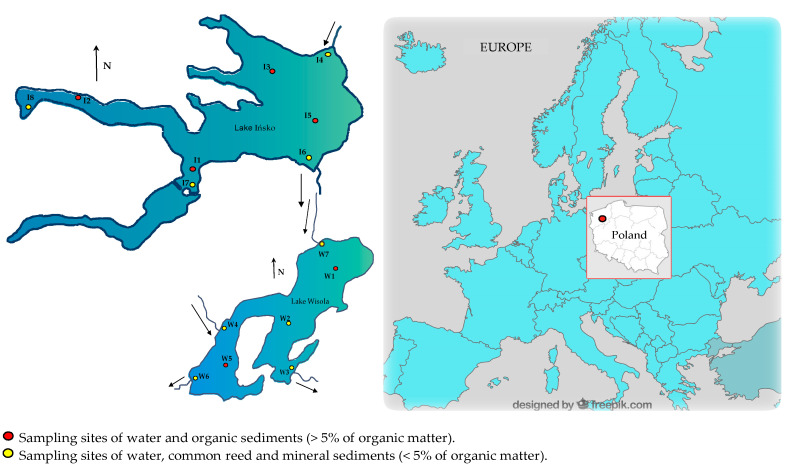
Location of the sampling sites for abiotic and biotic components of lakes Ińsko and Wisola (north-western Poland).

**Figure 2 animals-13-00697-f002:**
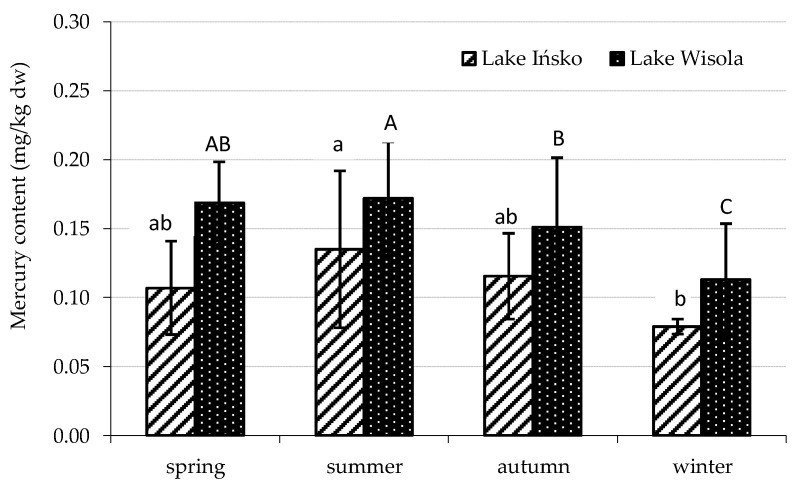
Seasonal variability in mercury content in organic sediments of lakes Ińsko and Wisola. Statistically significant seasonal differences in the content of Hg in the sediments of Lake Ińsko are indicated by different lowercase letters, and in sediments of Lake Wisola by different capital letters (Duncan’s test, *p* < 0.05).

**Table 1 animals-13-00697-t001:** Accuracy of the methods applied in the study.

Reference Materials	Total Hg, µg/kg (Mean ± SD)		Recovery, %	RSD, %
	Certified Value	Obtained Value		
MESS-3	0.091 ± 0.009	0.093 ± 0.006	102.2	6.4
INCT-MPH-2	0.018 ± 0.002	0.019 ± 0.002	105.5	10.5
DOLT-2	2.14 ± 0.28	2.16 ± 0.13	100.9	6.0
Fish-Paste-2	0.10 ± 0.07	0.095 ± 0.009	95	9.5

SD, standard deviation; *n* = 3.

**Table 2 animals-13-00697-t002:** Mercury concentration and other parameters of water from lakes Ińsko and Wisola (mean ± SD).

Lake	Water *	N	Hg (ng/L)	Transparency/ Depth (m) **	Temperature (°C)	pH	O_2_ (mg/L)
Ińsko	surface layer	40	1.2 ± 0.6 (0.5–2.7)	5.1 ± 0.6 (4.0–6.0)	12.1 ± 7.1 (7.1–21.0)	8.0 ± 0.3 (7.3–8.8)	9.0 ± 1.7 (6.5–12.6)
	near-bottom layer	16	1.1 ± 0.4 (0.6–2.1)	9.7 ± 2.2 (7.3–14.0)	8.8 ± 5.2 (3.4–18.6)	7.9 ± 0.5 (7.1–9.2)	7.4 ± 1.9 (3.6–10.0)
	tributaries	5	1.3 ± 0.5 (0.8–1.9)	-	11.8 ± 7.4 (3.2–20.7)	7.2 ± 0.2 (6.9–7.6)	7.4 ± 2.0 (6.1–10.0)
	outflows	3	0.9 ± 0.02 (0.8–1.1)	-	10.9 ± 10.1 (3.7–18.0)	8.1 ± 0.1 (8.0–8.2)	8.7 ± 0.2 (8.5–8.8)
Wisola	surface layer	24	1.5 ± 1.1 (0.4–4.7)	2.6 ± 1.2 (1.5–5.5)	12.5 ± 7.3 (3.7–21.0)	8.3 ± 0.6 (7.4–9.4)	9.6 ± 1.3 (7.6–12.5)
	near-bottom layer	24	1.6 ± 1.3 (0.7–5.0)	9.8 ± 2.5 (6.0–14.0)	7.9 ± 3.7 (4.2–15.3)	7.6 ± 0.5 (7.1–8.9)	4.3 ± 3.8 (0.2–10.6)
	tributaries	12	2.2 ± 1.4 (0.9–5.4)	-	11.8 ± 6.5 (1.6–19.4)	7.5±0.4 (6.8–7.9)	7.1 ± 2.8 (2.0–10.4)
	outflows	13	1.5 ± 1.2 (0.4–4.3)	-	13.0 ± 6.9 (3.5–19.9)	8.2 ± 0.8 (6.4–9.2)	7.7 ± 3.2 (2.8–12.8)

N, number of samples taken; * the tributaries and outflows dried up periodically; ** for surface waters, transparency was determined by the Secchi disk test, and for bottom waters, tributaries and outflows, the depth from which the sample was taken is given.

**Table 3 animals-13-00697-t003:** Mercury content and other parameters of bottom sediments from lakes Ińsko and Wisola (mean ± SD; min–max).

Lake	Bottom Sediments	N	Hg (mg/kg dw)	Depth (m)	pH	Organic Matter (%)	Accumulation Factors (*AF*)
Ińsko	Organic * (muddy)	48	0.109 ± 0.037 (0.034–0.352)	10.4 ± 2.1 (7.5–13.0)	6.7 ± 0.2 (6.5–7.2)	20.2 ± 3.5 (13.6–28.1)	94,783 ± 32,174 (29,565–306,087)
	Mineral ** (sandy)	72	0.014 ± 0.006 (0.003–0.055)	7.2 ± 1.1 (6.0–9.0)	6.8 ± 0.4 (6.2–7.5)	2.8 ± 1.9 (0.8–4.7)	12,174 ± 5217 (2608–47,826)
Wisola	Organic (muddy)	48	0.148 ± 0.047 (0.053–0.244)	10.6 ± 2.4 (7.5–14.0)	6.6 ± 0.3 (6.4–7.7)	37.1 ± 6.9 (22.5–47.9)	95,484 ± 30,323 (34,194–157,419)
	Mineral (sandy)	120	0.019 ± 0.020 (nd–0.078)	7.2 ± 1.1 (6.0–9.0)	6.8 ± 0.4 (6.2–7.5)	2.8 ± 1.9 (0.9–4.7)	12,258 ± 12,903 (nd–50,323)

* organic matter > 5% from sampling sites I3, I5 (Lake Ińsko) and W1, W5 (Lake Wisola); ** organic matter < 5% from sampling sites I4, I7, I8 (Lake Ińsko) and W2, W3, W4, W6, W7 (Lake Wisola) (Figure 1); N, number of samples; nd, not detectable.

**Table 4 animals-13-00697-t004:** Mercury content and bioaccumulation factors in organs of common reed (mean ± SD; min–max) *****.

Organs	Lake Ińsko			Lake Wisola		
	Hg (mg/kg dw)	*BAF_w_*	*BAF_s_*	Hg (mg/kg dw)	*BAF_w_*	*BAF_s_*
leaves	0.010 ^aA^ ± 0.005 (0.001–0.020)	9386 ± 4032 (6364–9386)	0.92 ± 0.68 (0.26–2.00)	0.013 ^aA^ ± 0.008 (0.003–0.037)	11,302 ± 5546 (5541–11,302)	0.89 ± 0.68 (0.66–1.46)
stems	0.009 ^aA^ ± 0.010 (0.001–0.045)	7870 ± 7017 (1744–21,885)	0.63 ± 0.38 (0.22–1.17)	0.008 ^bA^ ± 0.007 (0.001–0.036)	6350 ± 4680 (1079–14,286)	0.55 ± 0.40 (0.11–1.30)
rhizome	0.006 ^bA^ ± 0.004 (0.001–0.018)	4696 ± 2458 (1310–8529)	0.39 ± 0.12 (0.17–0.51)	0.007 ^bA^ ± 0.012 (0.001–0.099)	6361 ± 5771 (869–18,695)	0.57 ± 0.42 (0.07–1.26)
roots	0.013 ^aA^ ± 0.013 (0.001–0.058)	10,558 ± 7964 (3251–25,330)	0.88 ± 0.46 (0.41–1.51)	0.017 ^cB^ ± 0.012 (0.001–0.065)	13,120 ± 6331 (4668–23,901)	1.35 ± 0.97 (0.36–2.79)

* Different small letters next to the values indicate significant differences within the column; different capital letters indicate significant differences in Hg bioaccumulation by the same organs between lakes (Duncan’s test, *p* < 0.05); N = 72; *BAF_w_*, bioaccumultion factor from water; *BAF_s_*, bioaccumultion factor from sediments.

**Table 5 animals-13-00697-t005:** Mercury concentration in organs and food content of fish from Ińsko and Wisola lakes (mg/kg ww; mean ± SD; min–max).

Organs	Lake Ińsko			Lake Wisola		
	Pike (*n* = 80)	Bream (*n* = 80)	Roach (*n* = 80) *	Pike (*n* = 80)	Bream (*n* = 80)	Roach (*n* = 80)
gonads	0.020 ± 0.017 ^a^ (nd–0.077)	0.007 ± 0.004 ^b^ (0.001–0.022)	0.015 ± 0.012 ^a^ (0.004–0.065)	0.025 ± 0.023 ^a^ (0.005–0.115)	0.009 ± 0.007 ^b^ (0.001–0.039)	0.016 ± 0.007 ^b^ (0.007–0.03)
kidneys	0.079 ± 0.022 ^a^ (0.047–0.15)	0.014 ± 0.007 ^bA^ (0.005–0.035)	0.027 ± 0.008 ^c^ (0.015–0.041)	0.086 ± 0.035 ^a^ (0.035–0.164)	0.011 ± 0.005 ^bB^ (0.004–0.026)	0.032 ± 0.011 ^c^ (0.013–0.064)
liver	0.07 ± 0.024 ^a^ (0.029–0.159)	0.021 ± 0.010 ^b^ (0.005–0.041)	0.031 ± 0.011 ^b^ (0.014–0.052)	0.079 ± 0.038 ^a^ (0.032–0.148)	0.018 ± 0.013 ^b^ (0.006–0.05)	0.043 ± 0.021 ^c^ (0.012–0.08)
spleen	0.06 ± 0.061 ^a^ (0.018–0.36)	0.012 ± 0.006 ^b^ (0.005–0.024)	0.02 ± 0.007 ^bA^ (0.004–0.03)	0.049 ± 0.014 ^a^ (0.031–0.077)	0.011 ± 0.004 ^b^ (0.007–0.018)	0.042 ± 0.014 ^aB^ (0.015–0.059)
stomach	0.064 ± 0.067 ^a^ (nd–0.46)	0.014 ± 0.012 ^bA^ (0.007–0.09)	-	0.065 ± 0.024 ^a^ (0.015–0.114)	0.009 ± 0.004 ^bB^ (0.003–0.018)	-
intestine	0.068 ± 0.022 ^a^ (0.021–0.12)	0.011 ± 0.004 ^bA^ (0.004–0.021)	0.03 ± 0.011 ^c^ (0.011–0.056)	0.072 ± 0.026 ^a^ (0.037–0.132)	0.007 ± 0.003 ^bB^ (0.001–0.012)	0.029 ± 0.014 ^c^ (0.011–0.059)
food content	0.071 ± 0.021 ^a^ (0.04–0.122)	0.022 ± 0.010 ^b^ (0.007–0.054)	0.037 ± 0.019 ^b^ (0.004–0.06)	0.077 ± 0.026 ^a^ (0.025–0.125)	0.016 ± 0.012 ^b^ (0.006–0.05)	0.035 ± 0.019 ^c^ (0.005–0.081)
skin	0.004 ± 0.003 ^a^ (nd-0.009)	0.007 ± 0.004 ^a^ (0.001–0.022)	0.026 ± 0.015 ^bA^ (0.003–0.054)	0.003 ± 0.002 ^a^ (0.001–0.006)	0.006 ± 0.006 ^b^ (nd-0.02)	0.014 ± 0.014 ^cB^ (0.001–0.042)
gill	0.041 ± 0.013 ^aA^ (0.022–0.069)	0.006 ± 0.002 ^b^ (0.002–0.013)	0.019 ± 0.007 ^b^ (0.008–0.03)	0.067 ± 0.071 ^aB^ 0.025–0.42	0.006 ± 0.003 ^b^ (nd–0.012)	0.023 ± 0.007 ^b^ (0.01–0.035)
muscleTHg **	0.008 ± 0.004 ^aA^ (0.003–0.022)	0.005 ± 0.003 ^bA^ (0.001–0.011)	0.051 ± 0.028 ^c^ (0.011–0.116)	0.034 ± 0.064 ^aB^ (0.002–0.214)	0.013 ± 0.013 ^bB^ (0.001–0.045)	0.067 ± 0.056 ^c^ (0.002–0.155)
MeHg ***	0.007 (0.003–0.018)	0.004 (0.001–0.009)	0.043 (0.009–0.097)	0.028 (0.002–0.178)	0.011 (0.001–0.038)	0.056 (0.002–0.129)

* The number of individuals was 80, but the number of samples of some organs was lower due to insufficient weight of the organ for analysis (e.g., spleen). In such cases, organs from fish of similar size were combined to obtain the appropriate weight. ** THg also expresses 100% MeHg share, assumption according to EFSA [41]; *** MeHg, mercury converted to methyl form [42]; nd, not detectable; Within a given lake, different lower case letters in the same rows indicate significant inter-species differences in mercury accumulation by the same organ. Different capital letters indicate significant differences in Hg accumulation within the same species from the two lakes (Duncan’s test, *p* < 0.05).

**Table 6 animals-13-00697-t006:** Bioaccumulation factors (*BAF*).

Species	Organs *	Lake Ińsko				Lake Wisola			
		Mean	±SD	Min	Max	Mean	±SD	Min	Max
pike	digestive tract	57,411	11,361	40,924	76,060	59,914	29,265	11,525	103,778
	skin	3478	1612	1134	5487	2511	1334	280	4748
	gills	36,932	9780	22,954	51,927	42,511	18,183	19,447	76,668
bream	digestive tract	10,737	1781	8362	13,496	6640	2982	1901	9636
	skin	5490	3516	1180	12,400	4880	6173	979	17,325
	gills	5227	1222	2984	6344	4728	2351	1677	8077
roach	digestive tract	26,169	7242	13,194	37,356	24,339	10,757	8414	38,871
	skin	23,084	13,709	4439	47,561	22,564	14,092	1447	46,516
	gills	16,052	4852	10,125	23,279	15,079	6778	3919	23,671

* The table presents calculations only for organs representing the main routes of exposure of fish to Hg in the aquatic environment.

**Table 7 animals-13-00697-t007:** Linear correlation coefficient (r) between mercury content in fish organs and fish size (*p* < 0.05).

Species	Organs	Correlation Coefficient (r)	
		Lake Ińsko	Lake Wisola
bream	gonads	length (0.69) weight (0.77)	
	spleen	length (0.62)	length (0.69) weight (0.66)
	muscles	length (0.78) weight (0.81)	length (0.61) weight (0.76)
	stomach		weight (0.43)
roach	spleen	length (0.67) weight (0.72)	
	kidneys		length (−0.44) weight (−0.41)
	gills		length (0.71) weight (0.59)

*p*, significance level.

**Table 8 animals-13-00697-t008:** Potential human health risk assessment for adults (mean, min–max).

Lake	Species	Calculations for THg *				Calculations for MeHg **			
		EDI (µg/kg bw)	EWI (µg/kg bw)	% of TWI	THQ	EDI (µg/kg bw)	EWI (µg/kg bw)	% of TWI	THQ
Ińsko	pike	0.005 (0.002–0.013)	0.34 (0.013–0.094)	0.038 (0.014–0.104)	0.049 (0.018–0.135)	0.004 (0.002–0.011)	0.029 (0.011–0.079)	0.031 (0.012–0.086)	0.041 (0.015–0.112)
	bream	0.003 (0.001–0.007	0.021 (0.004–0.047)	0.024 (0.005–0.052)	0.031 (0.006–0.067)	0.003 (0.001–0.006)	0.018 (0.004–0.039)	0.020 (0.004–0.043)	0.026 (0.005–0.056)
	roach	0.031 (0.007–0.071)	0.219 (0.047–0.498)	0.240 (0.052–0.547)	0.313 (0.067–0.711)	0.026 (0.006–0.059)	0.189 (0.039–0.415)	0.20 (0.043–0.456)	0.260 (0.056–0.592)
Wisola	pike	0.021 (0.001–0.131)	0.146 (0.009–0.918)	0.160 (0.009–1.009)	0.208 (0.012–1.312)	0.017 (0.001–0.109)	0.122 (0.007–0.765)	0.134 (0.008–0.841)	0.174 (0.010–1.093)
	bream	0.008 (0.001–0.028)	0.056 (0.004–0.193)	0.061 (0.005–0212)	0.080 (0.006–0.276)	0.007 (0.001–0.023)	0.046 (0.004–0.161)	0.051 (0.004–0.177)	0.066 (0.005–0.230)
	roach	0.041 (0.001–0.095)	0.287 (0.009–0.665)	0.316 (0.009–0.731)	0.411 (0.012–0.950)	0.034 (0.001–0.079)	0.240 (0.007–0.554)	0.263 (0.008–0.609)	0.342 (0.010–0.792)

* THg, total Hg in fish muscles (value corresponding to 100% share of MeHg in fish muscles according to EFSA [41]); ** MeHg, calculated methylmercury [42]; Calculations were made for an adult with a body mass of 70 kg; EDI, estimated daily intake; THQ, target hazard quotient; TWI, tolerable weekly intake.

## Data Availability

The data that support the findings of this study are available from the corresponding author, upon reasonable request.

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
