# Peer review of "Mercury in Selected Abiotic and Biotic Elements in Two Lakes in Poland: Implications for Environmental Protection and Food Safety"

_animals, 2023, doi:10.3390/ani13040697_

Round 1
Reviewer 1 Report
The manuscript animals-2168363 titled "Mercury in selected abiotic and biotic elements of Ińsko and Wisola lakes (north-western Poland): implications for environmental protection and food and consumer safety" by Monika Rajkowska-Myśliwiec, Mikołaj Protasowicki is an environmental and toxicological study on Mercury (Hg) accumulation.
Bioaccumulation of mercury in aquatic food webs is an emerging problem for the human health risk (consumers of fish) and for the wildlife. In this study are shown data on Hg levels in biotic and abiotic elements of two lakes in Poland.
Although the results, the figures and the tables provided are clear and complete, the meaning and the real aim of this study is not clear. In the Discussion and the conclusion is evident the importance to remark as the levels of Mercury found in the lakes’ water were within the range characteristic of unpolluted waters and to enhance the consumer safety.
The results obtained from biotic and abiotic elements should be analysed underlying the scientific side of the data, providing a complete characterization of mercury level in the two polish lakes.
Here the list of suggestion and comments:
1. The title could be: “Mercury in selected abiotic and biotic elements of two lakes of Poland”, or also change the title highlighting the zoological contribute (considering also the topics of journal).
2. Lane 35: define the acronyms of THg and MeHg.
3. In the abstract it would be better to avoid inserting the specific values of the data (please, insert in the results only). In the abstract is important to give the aim of the study and also the most important results.
4. Lane 339 (Results): it would be appreciated a new table with all values of the Hg levels in the organic sediments from two lakes analysed in the different seasons. The authors should give a possible explanation of the seasonal difference of Hg levels.
5. Lane 467-469: this sentence it is not clear.
6. Lane 512-513: Decreasing Hg level in many organs of different species are already reported in the results; please insert it in the results or in the discussion only.
7. The muscles of roaches have a higher content mercury than pike or bream. The authors could provide a hypothesis to explain this difference. Moreover, the study suggest an interesting lack of biomagnification (lane 542), in the relationship between planktivorous, benthivores and predatory fishes. Can this be correlated with the potential environmental pollution or with specific differences of the habitat of two lakes?
8. Lane 574-575: in the table 7, values of THQ for the muscles of Pikes or the roaches from two lakes are not reported. Please, could you provide the missing data?
9. In Materials and methods, it is reported that an equal number of male and female fishes were collected. It was present a statistically significant difference for the Hg levels in both sexes?
10. Why the concentrations of Hg in water and bottom sediments of lake Ińsko were lower than in lake Wisola? Please, provide some possible causes to justify these differences.
These integrations are needed to improve the quality of manuscript.
Reviewer 2 Report
Dear Authors,
The manuscript is well-written, however, is very detailed and does not contain graphical elements as figures and/or diagrams that show trends, comparison of concentrations etc.
What is really new in the study? Please, show and explain better results of your detailed analysis. In the present form the manucript seems to be a report based on the laboratory analysis rather than scientific paper.
Minor chages:
L: 564- Lack of the percentage share
Reviewer 3 Report
The author examine Hg content in both organisms and environmental media from two lakes in north-western Poland. The results presented that Hg in water and bottom sediments of lake Ińsko were lower than in lake Wisola. Hg content and organic carbon were positive correlated. Hg concentration in organs of common reed from both lakes ranged from 0.006 to 0.017 mg/kg dw and its distribution in organs can be represented as: root > leaves > stems > rhizome. Muscles and skin of roach obtained high value of Hg. Health risks of organisms were evaluated in this work.
Major concern:
The author conducted lots of work to determine the overall Hg pollution in study areas. However, such many results can not investigate the underlying processes of Hg in environment, these likes to exhibit the pollution profiles of Hg in study areas. I suggest the author could further discuss some of data in this work. As well as, figures are needed for research paper. Lacking of figures makes this paper seems like a environmental report.
Minor concern:
Line 52-54: The toxicity of MeHg need to be described here.
Line 60: Which of trace elements?
Line 62: I suggested the author pay attention to Hg, rather than heavy metal.
Line 189: Hg in water sample is low. Which of vessel was used for collecting water samples? and if there are any procedures for eliminate the Hg in the vessel?
Line 198-200: What about the concentration of blank samples for water?
Line 250: ng/L
Line 317: Attention to the unit.
Line 473: More references need to be cited here.
Line 481-483: This statement need to be verified with MeHg value in sediment, otherwise, the author can not propose that there is a high methylation rate in study areas. In fact, the long period of growth for fish may generally elevate the Hg levels in fish. This paragraph need to be modified.
Reviewer 4 Report
GENERAL COMMENTS
The manuscript entitled: "Mercury in selected abiotic and biotic elements of Ińsko and Wisola lakes (north-western Poland): implications for environ-3 mental protection and food and consumer safety" provides multidisciplinary research on mercury accumulation at two Polish lakes. The article provides data from multiple set of biotic and abiotic parameters to determine interspecies comparisons and between-lakes comparisons. The authors concluded that no significant health risk is observed in both lakes. Interesting work with multiscale analysis. The manuscript needs some good re-editing in terms of the English language and a more compact and to the point discussion with emphasis on the ecological aspect of the research.
Title
Title needs to be rephrased. I recommend you change it to: “Mercury in selected abiotic and biotic elements of Ińsko and Wisola lakes (north-western Poland): implications for environmental protection and food consumption safety”
Simple Summary
L10-13: Please rephrase better the sentence – maybe breaking it in two phrases
L12: I don’t see the impact of fish length and weight through out the manuscript. Please be more specific
Abstract
L18: rephrase like this: “Mercury, which ... webs, poses a potential health risk to wildlife and to consumers of .. “
L19: I miss a sentence here linking the 1st sentence of the paragraph with the 2nd..
L25: better here and for the rest of the manuscript put units of measurements (for example here mg/kg) after the values depicted.
L28-29: Better rephrase
L35: please insert in brackets what THg and MeHg mean
Introduction
L41-43: Poor vocabulary and expression, the sentence needs rephrase. Please note that words like “enriched” are only used in a good way.
L43: Erase comma after that
L50: replace for with of
L50-52: please rephrase better
L52-54: please rephrase better
L55-56: Why?? Maybe insert a phrase here explaining why this is happening?
L61-63: “are useful not only..” please rephrase better
L65: “present conditions and changes .. “ – please be more specific
L68: “factors” please be more specific
L69-70: please rephrase (wrong use of the word “efficient” – likewise, it has a good meaning)
L73-75: I miss here the ecological importance of fish especially with regards to their trophic levels. More specifically, authors could discuss the ecological meaning here of being either a planktivorous fish, a benthic fish or a predator. In general, I feel that this linkage is missing from the whole manuscript: to discuss why these results appeared in relation to the fish trophical status.
L75: please replace the reference with a review or a reference more universal for fish toxicity.
L83-85: please rephrase better!
L89: “what is more” doesn’t fit there. Link in a better way fish organs with human nutrition
L89-91: needs rephrase
L93: please add a reference on the sentence “ Ingestion of low doses of MeHg may cause de-92 velopmental delays or other neurological problems”
L93: add author names istead of ref No
L99: erase alike
L99: “to some form of mercury” – please rephrase
L99: Replace “Holmes, 2009” with Ref No
L101: rephrase like this: “.. and sediments), biotic elements (common reed (Phragmites australis) …”
L103-106: Poor vocabulary and expression, please rephrase better!
L107-108: please erase “for the sake of their safety” or better rephrase
Materials and Methods
L112: Erase “the” other
L114: Here seems to appear a problem with the references number which keeps up till the end.
L137: Erase “The” research material
L138: erase “The” tested material
L139-140: “ and 139 fishes present at different trophic levels:” – poor English, please rephrase
L154: Delete “The” fish and add a comma after “under study”
L167: rephrase like this: “till analysis took place”.
L176: rephrase better
L190: rephrase as follows: “1 L of water samples was poured..”
L202-203: not sure if I am following here the symbols “-” in front of the values depicted. Please explain
L201-210: what about the fish different developmental stages? Did you take that into account?
L227-228: same here
Table 1: In all tables, it would be best to change the captions as follow: “SD, standard deviation; n=3” – try to avoid putiing this “-” symbol
L247-250: same here
L273: “For the purpose of risk assessment” à rephrase to “For risk assessment, ..”
L299-302: rephrase as shown: “ where: C, Hg content in fish (mg/kg ww); IR, fish ingestion rate (g/day); BW, body 299 weight (kg); EF, exposure frequency (365 days/year); ED, exposure duration (70 years); 300 RfD, oral reference dose (mg/kg/day) for Hg (1.0 × 10−4); AT, averaging exposure time 301 for non-carcinogens (365 days/year × ED)”
L331-333: the same
L341: put (mg/kg dw) after the values depicted
L347: Erase “Figure 1”
L355: rephrase like previous
L370-372: put mg/kg dm after depiction of the values
L377-378: rephrase like previous
L409: “with regards”
L429: put mg/kg body weight, after depiction of the values
L441-443: as previous
Discussion
L446: rephrase as follows: “.. (Table 2.), complying with ..”
L448-449: please rephrase
L461-462: please rephrase
L467-469: please rephrase
L496-500: At this part I miss the explanation on the leaves>stems>rhizome relationships with regards to ecology and Hg accumulation
L526-535: I miss here an explanation/assumption on why roach presented high levels of Hg in the skin, in contrast to the other species
L541-542: please explain in detail/expand this sentence. How is that evident here?
L541-548: In general, I miss in the discussion the more ecological part with regards to biomagnification and bioaccumulation and why this study contradicts the previous one.
L564: something missing here
L599-600: please rephrase better
Round 2
Reviewer 2 Report
no more comments
Reviewer 3 Report
The english language and style need to check before publication.
